# Perinatal Cerebral Ischemic Lesion and SARS-CoV-2 Infection during Pregnancy: A Case Report and a Review of the Literature

**DOI:** 10.3390/jcm11226827

**Published:** 2022-11-18

**Authors:** Claudia Brogna, Barbara Brogna, Margherita De Biase, Francesca Sini, Federica Mirra, Marianna Moro, Domenico M. Romeo

**Affiliations:** 1Pediatric Neurology Unit, Fondazione Policlinico Universitario “A. Gemelli”, IRCCS, 00168 Rome, Italy; 2Department of Radiology, “San Giuseppe Moscati” Hospital, Contrada Amoretta, 83100 Avellino, Italy; 3Pediatric Neurology Unit, Università Cattolica del Sacro Cuore, 00168 Rome, Italy

**Keywords:** ischemic lesion, neonate, SARS-CoV-2, pregnancy

## Abstract

Perinatal stroke is related to possible differences in predisposing factors and outcomes between acutely and retrospectively diagnosed cases. In most cases, there are different risk factors and infections that could play an important role. Thus far, different clinical manifestations have been reported in children presenting with severe acute respiratory syndrome coronavirus 2 (SARS-CoV-2), ranging from asymptomatic status to severe disease sustained by an immune-mediated inflammatory response. SARS-CoV-2 has been associated with severe neurological diseases including seizures and encephalitis in both adults and children. However, there are still few reports regarding the possible relation between SARS-CoV-2 infection of mothers during pregnancy and the neurologic outcome of the newborns. We described the case of a newborn diagnosed with a perinatal stroke, born at 35 weeks of gestation from a mother presenting with SARS- CoV-2 infection during the last months of pregnancy. We also added a brief review of the literature with similar cases. Close monitoring and early intervention in young children born to infected mothers would be highly recommended for the potential neurodevelopmental risk.

## 1. Introduction

Perinatal ischemic stroke is a cerebrovascular event occurring between the 20th week of pregnancy and the 28th day of postnatal life. The incidence of perinatal stroke is estimated to be from 63 per 100,000 (1 to 1587) to 1 per 2660 live births [1,2,3]. It is usually caused by vascular occlusion in an arterial territory secondary to arterial thrombosis or embolization and it usually involves the middle cerebral artery. 

The pathogenesis of perinatal brain injury is complex and multifactorial. It is not always possible to recognize the cause of perinatal stroke, and this could remain unclear. Independent risk factors for perinatal arterial stroke include the prolonged rupture of membranes, chorioamnionitis, preeclampsia and intrauterine growth restriction. The lipoprotein (a), factor V Leiden mutation, hyperhomocysteinemia, protein C deficiency or thrombophilia could also be considered. On the other hand, various noxious stimuli, including perinatal inflammation, chronic and acute hypoxia, hyperoxia, stress and drug exposure could contribute to the pathogenesis of the stroke associated with neuroinflammatory processes involving the peripheral immune system [4]. 

Although a possible association between SARS-CoV-2 infection and stroke in children has been reported, mainly in older children with comorbidities such as immunosuppression and a hypercoagulability state [5,6], few studies have described cases of just cerebral stroke in newborns born from mothers infected by SARS-CoV-2 during the pregnancy [7,8,9,10,11]. 

Different clinical manifestations related to SARS-CoV-2 are reported in children, ranging from an asymptomatic status to severe disease sustained by an immune-mediated inflammatory response, with or without associated multisystem-associated inflammatory syndrome in children (MIS-C), characterized by persistent fever, two or more organs’ involvement, elevated inflammatory markers and the exclusion of other diagnoses [12]. Singer et al. [13] described the different impacts of human coronavirus infections, including SARS-CoV-2, on a child’s nervous system. Children affected by non-SARS-CoV-2 infections have reported acute flaccid paralysis, acute disseminated encephalomyelitis, encephalitis and seizures, whereas cerebrovascular events (ischemic, hemorrhagic and microvascular strokes) were reported most often in association with acute SARS-CoV-2 infection.

Thus far, several studies have tried to clarify the SARS-CoV-2 transmission mechanisms and, in particular, the impact of infection on pregnant women and their babies [14,15,16]. SARS-CoV-2 infections in newborns have usually been considered to be acquired by contact during or after birth and vertical transmission has been suspected. 

Several epidemiological studies reported that SARS-CoV-2 infection may be associated with an increased risk of perinatal complications [17,18]. A strict association between placental infections induced by SARS-CoV-2 and fetal malperfusion due to an altered coagulative or microangiopathic state has been stated, resulting in an extensive placental damage, causing placental malperfusion and insufficiency that can lead to a hypoxic ischemic fetal demise or neonatal death [1,19,20].

However, there are still few reports in the literature regarding the neurological consequences of mothers’ SARS-CoV-2 infection on their babies during pregnancy. 

We describe the case of a newborn diagnosed with a perinatal stroke, born from a mother presenting with SARS-CoV-2 infection during the last months of pregnancy (written informed consent was obtained by both parents). We also report a recent review of the literature with similar clinical cases. 

Data acquisition and analysis was performed in compliance with protocols approved by the Ethical Committee of Fondazione Policlinico Gemelli Hospital. (Ethical approval ID number 3544).

## 2. Case Report 

A second-child male baby was born at 35 weeks gestation with a birth weight of 3380 gr from urgent cesarean section. His pregnancy was characterized by placental detachment on the second month of gestation. No clinical signs of diabetes or maternal hypertension were reported. During the second trimester the mother presented with cough, headache, chest pain and nausea. For this reason, she was taken to the emergency room of a local hospital and she resulted as positive to SARS-CoV-2 infection, detected by real-time reverse transcription polymerase chain reaction test (RT-PCR) on nasopharyngeal and oropharyngeal (NP/OP) swabs (gene N2 and E). She had not been vaccinated yet for SARS-CoV-2. She refused to perform arterial gas analysis, but her oxygen saturation was 99%. Blood tests showed a normal level of inflammatory markers and D-dimer value (Table 1). According to her previous history of pneumonia in the past (3 years before), a chest computed tomography (CT) was carried out showing no lung alterations. Clinical signs, treated with paracetamol only, continued for 1 week and she reported negative results of SARS-CoV-2 infection on nasopharyngeal and oropharyngeal (NP/OP) swabs after two weeks. Ultrasonography studies during the last weeks of pregnancy were reported as normal.

At birth, the newborn presented with respiratory distress (APGAR score 6^1^′–8^5^′) requiring respiratory assistance with a positive pressure mask and FiO_2_ 0.30, with admission to the Neonatal Intensive Care Unit. Cord/neonate blood gas analyses was normal. No skin rashes were observed. At day one, the newborn was lethargic, with a poor feeding attitude, and he presented with hypotonia and hyporeactivity, and apnea during the night. Instrumental investigations were then carried out. Complete blood count, C-reactive protein, electrolytes, blood glucose, liver and renal function studies were normal. RT-PCR for SARS-CoV-2 resulted as negative. Brain ultrasound showed suspected infarction involving the right basal ganglia (caudate nucleus and globus pallidus), ipsilateral internal capsule and increased periventricular echogenicity and near-normal cerebral blood flow (mild vasodilation). Because of the brain ultrasound findings, on the 2nd day, coagulation studies were carried out with normal results but for the D-Dimer value (Table 2). 

On the 5th day, a brain magnetic resonance imaging (MRI) study was obtained (Figure 1). It showed lesions of right frontal, temporal and parietal lobes consistent with infarcts in the right middle cerebral artery vascular territory with a modest hemorrhagic component of the right basal nucleus. The lesion had a cortico–subcortical and deep extension, with involvement in the caudate and lenticular nuclei and the internal and external capsule, the ventral part of the thalamus and mesencephalic and pontine right pyramidal pathway. Transthoracic echocardiography did not reveal an intracardiac thrombus. During his first weeks of life, the newborn showed a hypertonia on the left side of his body associated with poor spontaneous motility, more represented on the right side. Therefore, he started intensive physical therapy. The abdominal ultrasound was reported as normal. The child was evaluated every three months with a neurological assessment. At 12 months, the neurological assessment confirmed the clinical diagnosis of left hemiplegia with a prevalent involvement of the upper limb. The child did not show any seizures and the electroencephalogram (EEG) reported only a mild cerebral asymmetry.

Placental examination showed many thrombi in the chorionic vessels, as a fetal thrombotic vasculopathy. Thrombophilia screening and immunological blood tests on the mother reported as negative. Maternal history of varicella infection during her childhood was positive; screening for all neurological viral and bacterial infections, inborn errors of metabolism and mitochondrial gene abnormality-related diseases were performed with all resulting as normal.

## 3. Discussion 

Perinatal stroke usually manifests during the neonatal period with lethargy, seizures and apnea with respiratory distress. However, some infants can appear neurologically normal at this age, and perinatal stroke may be diagnosed later in the presence of clinical signs of hemiparesis or seizures. The presence of typical symptoms after birth leads the clinicians to perform neuroimaging, confirming the existence of an acute insult of vascular origin. 

In the last two years of the COVID-19 pandemic, few neonatal cases have been reported on neurological involvement related to a post-COVID systemic inflammatory response during pregnancy [7,21]. Some reports strongly suggest vertical transmission from the mother to the newborn through the placenta; however, perinatal and postnatal environmental exposures appear to be more common. Most newborns from positive mothers have resulted as negative, but some cases of positivity have been reported [14]. A recent retrospective observational study analyzed 688 babies born from 843 SARS-CoV-2 positive mothers in order to seek proof of vertical transmission. The systematic search resulted in a small number of cases of vertical transmission, suggesting that SARS-CoV-2 can be vertically transmitted, but the likelihood is poor [15]. In the cases of maternal-to-fetus virus transmission, it is unclear if transmission is transplacental, transcervical or related to environmental exposure. Findings of the ACE2 receptor on the placenta and the presence of SARS-CoV-2 mRNA and virions in syncytiotrophoblasts suggested a transplacental transmission. It could be mediated by direct damage to the villous tree with a break in the protective syncytiotrophoblast layer due to virus-induced apoptosis [16]. 

There are no specific clinical signs described for neonatal SARS-CoV-2 infections. In newborns, a severe central nervous system (CNS) involvement is reported as acute disseminated encephalomyelitis or with seizures, lethargy and documented ischemic lesions [8,9,13,14,15,16,22,23]. Reports of abnormal neuroimaging findings in children with SARS-CoV-2 infection resulted due to different mechanisms including virus-induced hyperinflammatory and hypercoagulable states, direct virus infection of the CNS and postinfectious immune-mediated processes [8,9,10,13,14,15,16,22].

The impact of a maternal inflammatory response following SARS-CoV-2 during pregnancy on cerebral neonate development has not yet been well investigated. In our case, SARS-CoV-2 infection appeared during the 3rd trimester of pregnancy, and the newborn, born at 35 gestational weeks, had an ischemic cerebral lesion associated to left hemiparesis and hypercoagulation status reported early with no evidence of positivity of SARS-CoV-2 infection at birth. 

Recent studies support the presence of neurological involvement in newborns tested as negative at birth [10,13,14,15,16,21,23], probably due to the fetus exposition to maternal infection, sustained inflammation and the consequent placental changes and prenatal immune activation. Activation of the maternal immune system exposes the fetus to a “cytokine storm”, with a massive release of proinflammatory mediator.

Pregnant women with SARS-CoV-2 infection have higher inflammatory cytokine IL-6 levels compared to non-pregnant women. Placental inflammation can cause fetal suffering and mortality due to the release of inflammatory cytokines into fetal blood and consequent fetal organ failure [24].

Thus far, a characteristic placental lesion related to SARS-CoV-2 infection has not been clearly demonstrated [25]. However, several recent studies reported that inflammation could induce vasculitis within the placenta, vascular endothelial damage, the exposure of thrombogenic basement membrane and the consequent activation of the clotting cascade leading to thrombotic events, decidual arteriopathy and maternal vascular malperfusion, with frequent cases of fetal suffering [7,8,9,10,11,19,20,22,26,27]. In our case, the finding of many thrombi in the placenta could be related to the pathogenic mechanism described.

In addition, a hyperreactive immune response to infection in pregnant women could trigger astroglia reactivity in the fetus brain, with sustained neuroinflammation. Both maternal and fetal inflammation could alter a newborn’s brain development and determine a risk for a poor neurodevelopmental outcome, even without a direct infection of the newborn [4,8,9,10,11,12,19,26,27,28,29,30,31]. An increased risk for disorders such as schizophrenia, autistic spectrum disorder (ASD) and attention-deficit/hyperactivity disorder (ADHD) [31,32,33] as a result of brain inflammation after SARS-CoV-2 infections has been recorded. However, the long-term impact on neurodevelopment deserves further investigation. 

Our case reported a clinical and radiological picture of perinatal ischemic stroke in a mother with a previous history of COVID-19 infection during pregnancy. It is not possible to confirm a direct link between SARS-CoV-2 infection and cerebral ischemic lesions; however, in the absence of other differential diagnoses, we cannot exclude the role of SARS-CoV-2 infection as a possible risk factor for brain lesions due to the hyperactive immune response being responsible for the placental thrombosis, as it has only recently been supposed from a recent survey about the possible relation between paediatric ischemic stroke and SARS-CoV-2 infection [9]. This report adds further information about the possible role of SARS-CoV-2 infection as a factor risk for the onset of ischemic brain lesions in neonates (Table 3).

## 4. Conclusions

This case and the cases reported in the literature highlight the importance of monitoring the newborn’s development in mothers affected by SARS-CoV-2 during the pregnancy. Further studies are needed to explain all the possible indirect mechanisms and the direct implications of SARS-CoV-2 infection on the brains of newborns in order to improve the assessment and management of the neurological outcomes and to reduce the morbidity. Close monitoring and early intervention in young children born from SARS-CoV-2-infected mothers would be highly recommended for the potential neurodevelopmental risk. The timeline of long-term dysfunction is still largely unclear. Further longitudinal studies are needed in order to better understand the natural history of these patients.

## Figures and Tables

**Figure 1 jcm-11-06827-f001:**
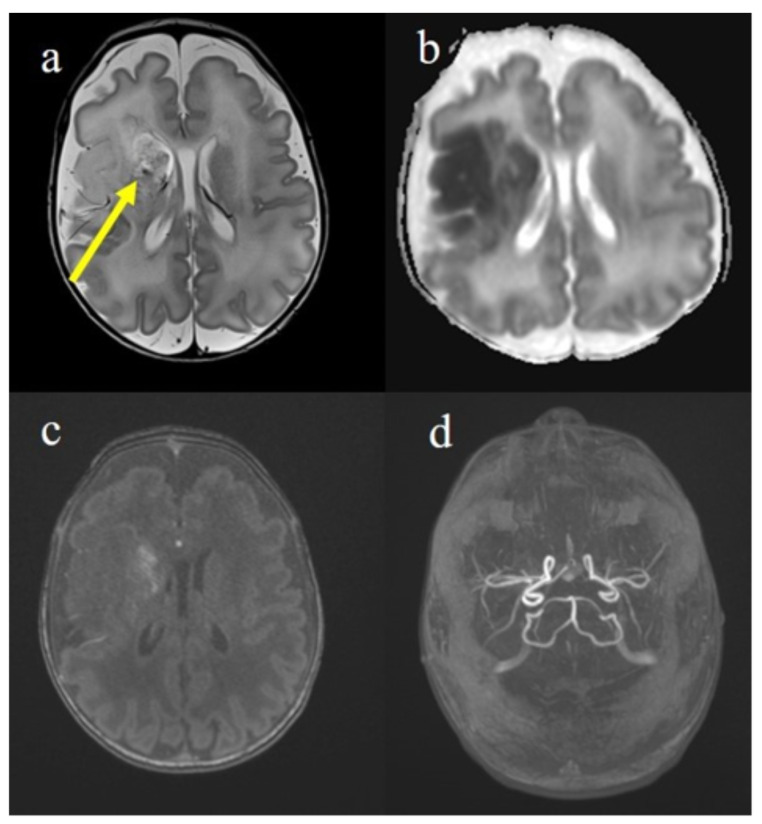
Brain MRI on 3 T performed on the 5th day of life showed hyperintense areas on the T2 sequence (**a**) in the right fronto-temporal regions involving the head, the caudate nucleus that appeared also with hypointense foci for hemorrhagic infiltration (yellow arrow). At the same level, areas with low ADC (**b**) corresponding to restricted diffusion were represented. On the Angio-TOF sequences, the petechial hemorrhagic infiltration was visible (**c**) in absence of arterial occlusion (**d**). These findings were compatible with ischemic stroke of the right middle cerebral artery (MCA) in a subacute phase.

**Table 1 jcm-11-06827-t001:** Laboratory findings of mother during SARS-CoV-2 infection.

Laboratory Findings	Value	Unit	Range
WBC	6.78	×10^3^/μL	4.00–10.00
RBC	4.69	×10^6^/μL	3.90–5.20
HGB	14.1	g/dL	12.0–16.0
HCT	43	%	38.0–46.0
MCV	91.9	fL	82.0–98.0
MCH	30.1	pg	27.0–34.0
MCHC	32.8	g/dL	32.0–36.0
PLT	389	×10^3^/μL	150–450
RDW	10.8	%	12.2–14.6
PDW	16.2	%	9.5–18.0
MPV	7.2	fL	6.0–12.3
PCT	0.28	%	0.120–0.425
%Neutrophil	70.2	%	40.0–75.0
%Lymphocyte	**16.6**	%	19.00–48.00
%Monocyte	**12.5**	%	<12.0
%Eosinophil	0.3	%	<6.0
%Basophil	0.4	%	<2.0
Neutrophil	4.8	×10^3^/μL	1.80–7.00
Lymphocyte	1.1	×10^3^/μL	1.00–4.80
Monocyte	0.8	×10^3^/μL	0.10–0.80
Eosinophil	0	×10^3^/μL	0.00–0.50
Basophil	0	×10^3^/μL	0.00–0.20
CRP	0.08	mg/dL	<0.5
Fibrinogen	225	mg/dL	200–400
D-Dimer	55	ng/mL	<280

In bold abnormal are reported abnormal values.

**Table 2 jcm-11-06827-t002:** Laboratory findings of the neonate at birth.

Laboratory Findings	Value	Unit	Range
PT	0.84	INR	0.8–1.2
PTT	27.1	sec.	24–40
Fibrinogen	207	mg/dL	150–450
D-Dimer	**1486**	ng/mL	<500
Antithrombin	71	%	39–87
Protein C Chromogenic	37.0	%	17–53
Protein S	105.3	%	60–140
Homocysteine	5.5	µmol/L	4.3–11.0
Lupus Anticoagulant	0.91	ratio	<1.2
Factor V	124.6	%	62–139
Anti-B2-glycoprotein IgG	<6.4	U/mL	<20
Anti-B2-glycoprotein IgM	<1.1	U/mL	<20

In bold are reported abnormal values.

**Table 3 jcm-11-06827-t003:** Neonatal ischemic brain lesions correlated with SARS-CoV-2 infection.

Authors	Demographics (Sex and Age)	MRI Brain Lesions	Possible Correlation with SARS-CoV-2
Vivanti et al., 2020 [11]	Male, preterm, gestational age 35+5 weeks;	At 11 days of life, bilateral gliosis of the deep white periventricular and subcortical matter, with slightly left predominance	Neonatal nasopharyngeal RT-PCR of SARS-CoV-2 at 1 h of life, at 3 and 18 days were all positive; cerebrospinal fluid (CSF) was negative for SARS-CoV-2. RT-PCR of SARS-CoV-2 in the mother was positive (E and S genes)
Brum et al., 2021 [7]	Male, born at term, 17 days	Two small ischemic focal lesions in the left frontal white matter with restriction on diffusion-weighted imaging	RT-PCR of SARS-CoV-2 positive in the neonate; mother RT-PCR of SARS-CoV-2 positive
Engert et al., 2021 [8]	Male, preterm, 33 weeks of gestation	On day 14 of life, bilateral intracranial bleedings in and outside the brain parenchyma with frontal accentuation	RT-PCR of SARS-CoV-2 after birth was negative in the mother (swab) and infant (swab and CSF). Maternal serology testing for SARS-CoV-2 (ELISA) revealed IgG antibodies against S1-protein and against N-protein
Beslow et al., 2021 [9]	Female, born at term, 4 days	Thalamo-capsular infarct	Neonatal RT-PCR of SARS-CoV-2 positive (2 days before stroke identified); mother positive for SARS-CoV-2.
Campi et al., 2022 [10]	Male, born at term, 4 days	Right thalamic infarction secondary to thrombosis of the internal cerebral veins, thrombosis of the Galen vein and venous sinuses confluence, posthemorrhagic dilation of the supratentorial ventricular system, and blood in the fourth ventricle	RT-PCR of SARS-CoV-2 negative in the neonate Mother had RT-PCR of SARS-CoV-2 positive at 33 weeks of gestation. Both mother and child had comparable anti-SARS-CoV-2 total antibody titer

## Data Availability

Data is contained within the article.

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
