# Peer review of "Perinatal Cerebral Ischemic Lesion and SARS-CoV-2 Infection during Pregnancy: A Case Report and a Review of the Literature"

_jcm, 2022, doi:10.3390/jcm11226827_

Round 1

Reviewer 1 Report (New Reviewer)

This is a very interesting case report of a newborn suffering from neonatal ischemic stroke after the mother had been diagnosed with SARS-CoV2 infection during pregnancy. Currently, there is little data regarding the deleterious effects of COVID on pregnant women and their newborns' central nervous system development. This paper can add to this growing database. In my opinion it is very well documented on the subject and thorough.

There are some minor recommendations/issues that must be addressed:

1. What was the vaccination status of the mother? I believe this should be mentioned as it could provide evidence to the efficacy (or lack thereof) of the vaccines in terms of stroke prevention in newborns. This is especially important, as the article by Beslow et al., which you cited, mentions that vaccines protect against childhood stroke, but there is scarce data on the protective effects of the mother being vaccinated.

2. The references in the tables do not match the provided reference list. Please correct this.

3. There are a few typos and grammatical errors within the manuscript. I'd recommend having the manuscript checked by a native English speaker.

4. What was/is the (planned) treatment and follow-up program for the child? It is mentioned that the child was seen again at twelve months, however what was the treatment regiment during that time and is there any follow-up or neurological treatment planned going forward?

Other than that, I commend the authors on their paper and wish them all the best in their endeavors!

Author Response

This is a very interesting case report of a newborn suffering from neonatal ischemic stroke after the mother had been diagnosed with SARS-CoV2 infection during pregnancy. Currently, there is little data regarding the deleterious effects of COVID on pregnant women and their newborns' central nervous system development. This paper can add to this growing database. In my opinion it is very well documented on the subject and thorough.

We thanks the reviewer for the comment

There are some minor recommendations/issues that must be addressed:

  1. What was the vaccination status of the mother? I believe this should be mentioned as it could provide evidence to the efficacy (or lack thereof) of the vaccines in terms of stroke prevention in newborns. This is especially important, as the article by Beslow et al., which you cited, mentions that vaccines protect against childhood stroke, but there is scarce data on the protective effects of the mother being vaccinated.

We thanks the riviewer for the observation. The mother had not been vaccinated yet for Covid-19. This has been added in the test.

  1. The references in the tables do not match the provided reference list. Please correct this.

We thanks the reviewer for the observation. These have been amended

  1. There are a few typos and grammatical errors within the manuscript. I'd recommend having the manuscript checked by a native English speaker.

Thanks, this has been done

  1. What was/is the (planned) treatment and follow-up program for the child? It is mentioned that the child was seen again at twelve months, however what was the treatment regiment during that time and is there any follow-up or neurological treatment planned going forward?

We thanks the reviewer for the observation. He early started intensive physical therapy and he has been evaluated every 3 months until 12 months and then every 6 months. This has been added in the test. 

Other than that, I commend the authors on their paper and wish them all the best in their endeavors!

Thanks very much.

Reviewer 2 Report (New Reviewer)

This is fairly well written paper on a relevant and interesting topic .

There are some grammatical and spelling errors through the manuscript that will need careful perusal .

I have the following comments and questions for the authors -

Obstetrical Details -

Abruption in the second month - this appears to be a a very early complications. Typically abruption is referred to in second half of pregnancy ? Can authors clarify ? 

Neonatal details :

Did the neonate have any clinical or electrical seizures ?  Any EEG results ? Was an abdominal ultrasound completed ?  

In the acute state - what was the extent of support needed - respiratory support ? How quickly could the feeds be established ?  Such details could enrich the case by giving an accurate estimation of disease severity ?

At 12 months - other than the motor development - status of other domains ? any epilepsy?

Discussion - Are the authors aware of any study on mothers positive with Sars CoV2 and placental pathology ? This could add to the discussion or be a potential future direction.

Author Response

This is fairly well written paper on a relevant and interesting topic .

We thanks the reviewer for the comment.

There are some grammatical and spelling errors through the manuscript that will need careful perusal .

I have the following comments and questions for the authors -

Obstetrical Details -

Abruption in the second month - this appears to be a a very early complications. Typically abruption is referred to in second half of pregnancy ? Can authors clarify ? 

 We thanks the reviewer for the observation. The mother presented a placental detachment during the second month of gestation. We change “abruption” with “detachment” in the test

Neonatal details :

Did the neonate have any clinical or electrical seizures ?  Any EEG results ? Was an abdominal ultrasound completed ?  

We thanks the reviewer for this observation. The neonate did not have any seizures, EEG showed only a mild asymmetria while abdnominal ultrasound was normal. This has been added in the test.

In the acute state - what was the extent of support needed - respiratory support ? How quickly could the feeds be established ?  Such details could enrich the case by giving an accurate estimation of disease severity ?

At birth the newborn presented with respiratory distress (APGAR score 61’-85’) requiring respiratory assistance with a positive pressure mask and FiO2 0.30, with admission to the Neonatal Intensive Care Unit. He did not have feed problem

At 12 months - other than the motor development - status of other domains ? any epilepsy?

At 12 months EEG showed only a mild cerebral asymmetry. This has been added in the test

Discussion - Are the authors aware of any study on mothers positive with Sars CoV2 and placental pathology ? This could add to the discussion or be a potential future direction.

Thanks the reviewer for this observation. Placental inflammation caused by Sars Cov 2 infection  can cause fetal suffering and mortality due to the release of inflammatory cytokines into fetal blood and consequent fetal organ failure. We have reported some references:

  1. Schwartz, D. A.; Avvad-Portari, E.; Babál, P.; Baldewijns, M.; Blomberg, M.; Zaigham, M. et al. Placental tissue destruction and insufficiency from COVID-19 causes stillbirth and neonatal death from hypoxic-ischemic injury: a study of 68 cases with SARS-CoV-2 placentitis from 12 countries. Arch Pathol Lab Med 2022, 146, 660-676.
  2. Menter, T.; Mertz, K. D.; Jiang, S.; Chen, H.; Monod, C.; Bruder, E. et al. Placental pathology findings during and after SARS-CoV-2 infection: features of villitis and malperfusion. Pathobiology 2021, 88, 69-77.

Several studies have already analysed the histology of the placentas from COVID-19 mothers, describing primarily microvascular changes, while an inflammatory response was occasionally encountered. To date, a characteristic placental lesion has not been clearly demonstrated, but most findings include features of maternal and fetal vascular malperfusion, which probably reflect the reduction in placental blood flow due to low oxygen level from the hypoxic respiratory disease and underlying hypercoagulable state induced by the COVID-19 infection.

We have added in the discussion these references

Pacu, I; RoÈ™u, G.A.; Zampieri, G.; Rîcu, A.; Matei, A.; DaviÈ›oiu, A.M.; Vlădescu, T.; Ionescu, C.A.SARS-CoV-2 Infection during Pregnancy and Histological Alterations in the Placenta. Diagnostics (Basel). 2022, 19,12:2258.

Shanes, E.D.; Mithal, L.B.; Otero, S.; Azad, H.A.; Miller, E.S.; Goldstein, J.A. Placental Pathology in COVID-19. Am. J. Clin. Pathol. 2020, 154, 23–32. 

Gao, L.; Ren, J.; Xu, L.; Ke, X.; Xiong, L.; Tian, X.; Fan, C.; Yan, H.; Yuan, J. Placental pathology of the third trimester pregnant women from COVID-19. Diagn. Pathol. 2021, 16, 8.

This manuscript is a resubmission of an earlier submission. The following is a list of the peer review reports and author responses from that submission.

Round 1

Reviewer 1 Report

This paper is a report discussing the complications of neonatal cerebral infarction and maternal coronavirus infection, this field is a topic that is currently trending.

However, it fails to demonstrate that neonatal cerebral infarction was caused by maternal coronavirus infection. Clinical data suggest a differential diagnosis of neonatal cerebral infarction, but it is only a diagnosis of exclusion.

The overall introduction and discussion are generalized and do not provide a scientific discussion focused on mechanism of this case relation between neonatal cerebral infarction and maternal corona virus infection.

MRI images should be presented as diffusion-weighted images (DWI) and ADC, STIR as well as 3T-MRA. And focus on the thalamic stria arteries should also be discussed the branch of cerebral middle artery. The authors wonder and speculate what and when week's gestation of the neonate had suffered the stroke based on the MRI lesion on this T2 image only.

Do the authors find a relationship between the number of weeks of infarction, as inferred from the relationship between MRI and maternal fetal heart rate/uterine contraction monitoring, which suggests that the neonate had an infarction episode?

Moreover, echocardiography should be performed trans-esophageally for precise evaluation.
Differential diagnosis may still be insufficient, including maternal history of varicella infection, PCR virus detaction in CSF,  IgG-index of spinal fluid, screening for inborn errors of metabolism, and mitochondrial gene abnormality-related diseases.

There is too little data presented to provide evidence that coronavirus caused neonatal cerebral infarction. It is still impossible to evaluate the paper as a paper that deduced and advocated a major conclusion from only one case report for publication.

Author Response

This paper is a report discussing the complications of neonatal cerebral infarction and maternal coronavirus infection, this field is a topic that is currently trending.

However, it fails to demonstrate that neonatal cerebral infarction was caused by maternal coronavirus infection. Clinical data suggest a differential diagnosis of neonatal cerebral infarction, but it is only a diagnosis of exclusion.

We thank the reviewer for the comment. The manuscript has been revised according to this consideration. Other infections have been excluded and thrombophilia and coagulation screening resulted normal. There was no other risk factors. Placenta The baby was born from an urgent cesarean section. Placental examination showed many thrombi in the chorionic vessels, as a fetal thrombotic vasculopathy. Covid-19 infection in the mother could be considered as a risk factor for perineonatal stroke by an indirect inflammatory mecchanism.

 The overall introduction and discussion are generalized and do not provide a scientific discussion focused on mechanism of this case relation between neonatal cerebral infarction and maternal corona virus infection.

The introduction has been implemented

MRI images should be presented as diffusion-weighted images (DWI) and ADC, STIR as well as 3T-MRA. And focus on the thalamic stria arteries should also be discussed the branch of cerebral middle artery. The authors wonder and speculate what and when week's gestation of the neonate had suffered the stroke based on the MRI lesion on this T2 image only.

We thank the reviewer for the valuable observations. We have added other sequence in order to describe the perinatal ischemic stroke in the subacute phase. Therefore, we reported the T2 sequence on the axial plane at the level of the caudate nucleus because at this level it better represented the subacute phase with petechial hemorrhagic infiltration. We reported the DWI ADC map. We also reported the MRI with angio-TOF sequences and with the 3D reconstruction that showed the restored flow of the right middle cerebral artery. The sequences are performed with a MRI at 3T that we have specified in the legend.

Do the authors find a relationship between the number of weeks of infarction, as inferred from the relationship between MRI and maternal fetal heart rate/uterine contraction monitoring, which suggests that the neonate had an infarction episode?

No, we did not found any correlation.

Moreover, echocardiography should be performed trans-esophageally for precise evaluation.

This information has been added

Differential diagnosis may still be insufficient, including maternal history of varicella infection, PCR virus detaction in CSF,  IgG-index of spinal fluid, screening for inborn errors of metabolism, and mitochondrial gene abnormality-related diseases.

These informations have been added

There is too little data presented to provide evidence that coronavirus caused neonatal cerebral infarction. It is still impossible to evaluate the paper as a paper that deduced and advocated a major conclusion from only one case report for publication.

We thank the reviewer fot the comment. The aim of this manuscript is to consider the possibile role of a later COVID-19 infection during pregnancy  on the onset of the perinatal stroke, as a risk factor and the importance to monitor mothers during pregnancy when affected by Sars COV2 infection. We specified it in the conclusions.

Reviewer 2 Report

The authors reported a case of neonate born from mother who was positive SARS-CoV-2 PCR during pregnancy presented with a perinatal stroke. The case report is meaningful because of the possible negative impact of COVID-19 infection during pregnancy on neurological outcome in affected infants.

Comment

1. The association between COVID-19 infection during pregnancy and a perinatal stroke is an important topic due to possible negative neurological outcome of baby. On the other hand, the incidence of perinatal stroke is estimated to happen 63 per 100,000 (1 to 1587) to 1 per 2,660 live birth according to an epidemiological study outbreak and a recent multicenter study before COVID-19 outbreak. (PMID: 17585082) (PMID: 35400054) As the authors mentioned in the report, epidemiological evaluation is necessary to prove the association between COVID-19 infection and perinatal stroke in the future. I guess readers want to know the difference between a perinatal stroke possibly related to COVID-19 infection and that caused by other etiology. Thus, I recommend the authors to focus on the difference or similarity of the perinatal stroke possibly related to COVID-19 infection and other etiology in the case report.

2. I recommend to add more detailed data on differential diagnosis that might result in a perinatal stroke other than COVID-19 infection including negative results.

2. More detailed data at birth is needed, such as vital signs, blood gas analysis and Apgar score.

Author Response

The authors reported a case of neonate born from mother who was positive SARS-CoV-2 PCR during pregnancy presented with a perinatal stroke. The case report is meaningful because of the possible negative impact of COVID-19 infection during pregnancy on neurological outcome in affected infants.

The association between COVID-19 infection during pregnancy and a perinatal stroke is an important topic due to possible negative neurological outcome of baby. On the other hand, the incidence of perinatal stroke is estimated to happen 63 per 100,000 (1 to 1587) to 1 per 2,660 live birth according to an epidemiological study outbreak and a recent multicenter study before COVID-19 outbreak. (PMID: 17585082) (PMID: 35400054) As the authors mentioned in the report, epidemiological evaluation is necessary to prove the association between COVID-19 infection and perinatal stroke in the future. I guess readers want to know the difference between a perinatal stroke possibly related to COVID-19 infection and that caused by other etiology. Thus, I recommend the authors to focus on the difference or similarity of the perinatal stroke possibly related to COVID-19 infection and other etiology in the case report.

We thanks the reviewer for the precious comment that help to improve di introduction. The aim of this manuscript is to consider the possibile role of a later COVID-19 infection during pregnancy  on the onset of the perinatal stroke, as a risk factor and the importance to monitor mothers during pregnancy when affected by Sars COV2 infection. This has been added in the introduction and in the discussion

I recommend to add more detailed data on differential diagnosis that might result in a perinatal stroke other than COVID-19 infection including negative results.

Thanks the reviewer for the comment. This information has been added.

More detailed data at birth is needed, such as vital signs, blood gas analysis and Apgar score

Thanks the reviewer for the comment. These informations have been added.

Round 2

Reviewer 1 Report

Thank you for changing the title, the introduction, the MRI comments, and for making the report more balanced overall.

However, the reviewers still consider this case to be a speculative case report and not a chemically based case report. However, the reviewers still consider this case to be speculative and not chemically based, because the direct causal role of coronavirus 19 as a causative mechanism of ischemic stroke lesions in neonates is unknown.

Similar speculation has been done with influenza virus, and ADHD in newborns of pregnant women with influenza virus, but the scientific evidence for this is still lacking.

Indeed, there are many cases of cerebral infarction in newborns for which the cause is unknown, and various factors have been discussed. Among them, the causal relationship of cerebral infarction after varicella virus infection has recently been discussed again based on pathological findings and antibody titers, but there are some negative opinions.

Of course I agree with you on the importance of monitoring pregnant women.

If this report were a report of three cases, I would consider accepting it, but if it is only a report of one case and published in the highly influential journal JCM, the reviewers will judge overall that the scientific evidence is poor.

Best regards,

Reviewer

Author Response

Thank you for changing the title, the introduction, the MRI comments, and for making the report more balanced overall.

However, the reviewers still consider this case to be a speculative case report and not a chemically based case report. However, the reviewers still consider this case to be speculative and not chemically based, because the direct causal role of coronavirus 19 as a causative mechanism of ischemic stroke lesions in neonates is unknown.

Similar speculation has been done with influenza virus, and ADHD in newborns of pregnant women with influenza virus, but the scientific evidence for this is still lacking.

Indeed, there are many cases of cerebral infarction in newborns for which the cause is unknown, and various factors have been discussed. Among them, the causal relationship of cerebral infarction after varicella virus infection has recently been discussed again based on pathological findings and antibody titers, but there are some negative opinions

We thank the reviewer for the comment. We report a clinical case presenting with a perinatal ischemic stroke during a cesarean delivery  withouth apparently other causes except for a previous mother COVID-19 infection; furthermore a placental examination showed many thrombi in the chorionic vessels, as a fetal thrombotic vasculopathy possibly related to the inflammation COVID related. A possible indirect effect of Covid-19 has been therefore supposed. This data has been also recently reported by Beslow LA et al (Ann Neurol. 2021.Pediatric Ischemic Stroke: An Infrequent Complication of SARS-CoV-2.) presenting a perinatal case with brain MRI lesions typical of stroke due to a possibile correlation with  mother SARS COV 2 Infection (see also Table 2 in word file). 

This has been added in the text, in the final part of discussion. We added also this reference.

“Our case reported a clinical and radiological picture of perinatal cerebral stroke in a mother with a previous history during pregnancy of Covid-19 infection. It is not possible to confirm a direct link between SARS -COV2 infection and cerebral ischemic lesions; however, in the absence of other differential diagnosis, we cannot exclude the role of COVID-19 infection as a possible risk factor for brain lesion due to the hyperactive immune response responsible of the placental thrombosis, as it has just been supposed from a recent survey about the relation between pediatric ischemic stroke and Sar Cov-2 infection [18]  “  

.

Of course I agree with you on the importance of monitoring pregnant women.

Thank you for this comment

If this report were a report of three cases, I would consider accepting it, but if it is only a report of one case and published in the highly influential journal JCM, the reviewers will judge overall that the scientific evidence is poor.

Our clinical case adds further evidence to the recent published data about the possibile relation between Sars cov2 infection and cerebral stroke in newborns.  It is an important epidemiological consideration for clinicians and for pregnants in order to improve the clinical history. This has been added in the text.

Reviewer 2 Report

Thank you for revising the manuscript. The authors answered all of my comments. The manuscript has improved with detailed discussion on differential diagnosis. I believe the case report deserves publication.

Author Response

We thanks the reviewer for the comment.